# Pharmacogenomics of Methotrexate Pathway in Rheumatoid Arthritis Patients: Approach toward Personalized Medicine

**DOI:** 10.3390/diagnostics12071560

**Published:** 2022-06-27

**Authors:** Hoda Y. Abdallah, Maha E. Ibrahim, Noha M. Abd El-Fadeal, Dina A. Ali, Gehad G. Elsehrawy, Rasha E. Badr, Howayda M. Hassoba

**Affiliations:** 1Medical Genetics Unit, Department of Histology and Cell Biology, Faculty of Medicine, Suez Canal University, Ismailia 41522, Egypt; hoda_ibrahim1@med.suez.edu.eg; 2Center of Excellence in Molecular & Cellular Medicine, Faculty of Medicine, Suez Canal University, Ismailia 41522, Egypt; noha_abdelfadeal@med.suez.edu.eg (N.M.A.E.-F.); dina_abdel-karim@med.suez.edu.eg (D.A.A.); 3Department of Physical Medicine, Rheumatology and Rehabilitation, Faculty of Medicine, Suez Canal University, Ismailia 41522, Egypt; maha.ibrahim@med.suez.edu.eg (M.E.I.); dr.ggrheu@gmail.com (G.G.E.); 4Department of Medical Biochemistry & Molecular Biology, Faculty of Medicine, Suez Canal University, Ismailia 41522, Egypt; 5Oncology Diagnostic Unit, Faculty of Medicine, Suez Canal University, Ismailia 41522, Egypt; 6Department of Clinical Pharmacology, Faculty of Medicine, Suez Canal University, Ismailia 41522, Egypt; 7Department of Clinical and Chemical Pathology, Faculty of Medicine, Port Said University, Port Said 42526, Egypt; dr.rashaemad@hotmail.com; 8Department of Clinical and Chemical Pathology, Faculty of Medicine, Suez Canal University, Ismailia 41522, Egypt

**Keywords:** rheumatoid arthritis, allelic discrimination, genotyping, methotrexate, single-nucleotide polymorphism, ATIC, GGH, DHFR, SLC19A1

## Abstract

Background: Methotrexate (MTX) is one of the most common medications used for rheumatoid arthritis (RA) treatment. Single-nucleotide polymorphisms (SNPs) could potentially predict variability in therapeutic outcomes. Aim: This study aims to assess the impact of SNPs in genes encoding for the MTX pathway for predicting clinical and therapeutic responses to MTX in a cohort of Egyptian patients with RA. Subjects and Methods: Data from 107 Egyptian RA patients (aged 44.4 ± 11.4 years) treated with MTX monotherapy, for a duration of 3.7 ± 3.3 years, were collected. Genotypes of 10 SNPs from four different genes were analyzed using the allelic discrimination PCR technique. Results: The *ATIC* rs3821353 G/T (*p* = 0.034) and the C/T and C/C of *SLC19A1* rs7279445 (*p* = 0.0018) were associated with a non-response to MTX, while *DHFR* rs10072026 C/T and C/C were associated with a good response (*p* < 0.001). Carriers of the *ATIC* rs382135 3 G (*p* = 0.001) and *ATIC* rs4673990 G (*p* < 0.001) alleles were more likely to develop RA, while the *SLC19A1* rs11702425 T (*p* < 0.001) and *GGH* rs12681874 T (*p* = 0.003) allele carriers were more likely to be protected against RA. Carriers of the *ATIC* rs4673990 A/G genotype (*p* < 0.001) were at risk of developing RA, while carriers of the following genotypes were mostly protected against RA: *ATIC* rs3821353 T/T (*p* < 0.001), *ATIC* rs3821353 G/G (*p* = 0.004), *SLC19A1* rs11702425 T/T (*p* = 0.001), *SLC19A1* rs11702425 C/T (*p* = 0.003), *GGH* rs12681874 C/T (*p* = 0.004) and *GGH* rs12681874 T/T (0.002). Conclusion: The genotyping of genes involved in the MTX pathway may be helpful to predict which RA patients will/will not benefit from MTX, and thus, may help to apply a personalized medicine approach in RA.

## 1. Background

Personalized Medicine is a recent field which aims to maximize the probability of therapeutic effectiveness and reduce the occurrence of adverse drug reactions [1]. The integration of the patient genome was considered an important factor responsible for intra- and interpatient drug therapeutic-outcome variability [2,3,4]. One of the major contributors to this development is pharmacogenomics [1].

Rheumatoid arthritis (RA) is considered a chronic systemic autoimmune disorder that affects mainly the lining of the synovial joints, causing destruction of the bone and cartilage, deformity progression, and finally, disability [5].

The prevalence of RA worldwide is somewhat consistent. It ranges from 0.5% to 1.1% [6]. The estimated prevalence of RA in Africa ranges from 0.06% to 3.4% in individual countries [7] and 0.29% in Egypt [8], with about 4.3 million Africans suffering from the disease [9]. A high prevalence of RA has been reported in the Pima Indians (5.3%) and the Chippewa Indians (6.8%), while a low prevalence has been reported in populations from China and Japan. These data support a genetic role in disease risk [10].

The introduction of modifying antirheumatic drugs had led to high efficacy in the treatment of RA. However, not all patients show the same degree of disease progression and response to these treatments. Of all the disease-modifying therapies, methotrexate (MTX), an antifolate agent, is basically considered a cornerstone drug for the treatment of RA [11]. Once it enters the cells, MTX is converted into methotrexate polyglutamates (MTXPGs) through the addition of glutamic acid residues in a sequential manner via the folylpolyglutamate synthetase (FPGS) enzyme [12].

MTX polyglutamation facilitates its intracellular retention, which promotes the inhibition of the methionine, folate, and adenosine pathways. Furthermore, it increases the de novo synthesis of pyrimidines and purines, which is critical for the antiproliferative and the anti-inflammatory therapeutic effects of MTX. Moreover, MTX is a low-priced and well-tolerated drug by most patients with RA [13].

On the other hand, MTX efficacy, based on the American College of Rheumatology criteria, is inconsistent (46–65%), with about one-third of patients not responding to MTX due to either adverse events (AEs) or lack of efficacy [14].

Thus far, no reliable clinical biomarkers have been identified to predict MTX efficacy or its toxic effects. A major challenge in human genetics is to devise a systematic strategy to integrate disease-associated variants with diverse genomic and biological datasets. This provides an insight into disease pathogenesis and guides the optimal treatment for complex traits such as RA. 

Therefore, there have been concerns about identifying novel screening tools that would constantly predict the efficacy and toxic effects of a disease-modifying antirheumatic drug (DMARD) before initiation in RA, wherein the pharmacogenetics field (PGx) plays a basic role [13]. PGx entails the study of the role of genetics in drug response, and in dealing with the influence of acquired and inherited genetic variation on drug response; this is achieved by correlating gene expression or single-nucleotide polymorphisms with drug pharmacokinetics, in addition to drug-receptor target effects [14].

Therapeutic response is a multi-factorial process in which multiple factors are involved, including disease-specific, individual, and genetic factors. The cross-talks of several coding genes involved in drug pharmacokinetics influence drug actions and, thus, these genes represent logical targets for pharmacogenetic testing to identify the predictors of response or toxicity. Over the past decade, numerous PGx studies have been undertaken to decipher the associations between SNPs in genes encoding enzymes in the cellular pathways that are influenced by MTX and the drug’s efficacy and toxicity [11,12,15].

It is currently well known that methotrexate transportation into enterocytes and liver cells occurs via various members in the solute carrier (SLC) family, e.g., SLC19A1. After the uptake, polyglutamation follows intracellularly via the enzymes FPGS and GGH, respectively. Active MTX-polyglutamates inhibit a variety of enzymes, including ATIC and DHFR, and have an indirect influence on MTHFR. Numerous SNPs in genes that are coding for MTX membrane transporters, P-glycoprotein [16,17], and folate transporter have been identified [18,19,20,21,22].

SLC19A1, DHFR, GGH, and ATIC are among the most-documented polymorphic gene loci coding for intracellular proteins related to the response to MTX in RA patients [22]. However, the observed results are not consistent owing to many reasons, such as diversity in the analyzed populations, MTX response criteria variations, and different sample sizes of the groups under study. Therefore, the goal of this study was to identify the potential association of multiple SNPs related to essential MTX pathway enzymes, such as ATIC, SLC19A1, GGH, and DHFR with MTX therapy response, in a cohort of Egyptian RA patients. Furthermore, we investigated the association of these SNPs in relation to RA disease risk via comparison to normal controls.

## 2. Subjects and Methods

### 2.1. Study Design and Population

This is an analytical cross-sectional study conducted between December 2019 and December 2021 on a cohort of 107 RA patients ≥ 18 years treated with MTX monotherapy for at least 3 months. One hundred age- and gender-matched healthy subjects served as a control group. All patients met the 2010 revised classification criteria of the American College of Rheumatology and the European League Against Rheumatism [23]. Patients were recruited from the Rheumatology and Rehabilitation Clinic, Suez Canal University (SCU) Hospital, Egypt. Laboratory investigations were performed in the Clinical Pathology laboratory of the SCU hospital, while the genetic analysis was performed at the Medical Genetics Unit in the Department of Histology and Cell Biology, and at the Center of Excellence of Molecular and Cellular Medicine in the Faculty of Medicine, SCU. Patients were excluded if they had a drug abuse history, recent pregnancy, or desire to become pregnant during the study. Additionally, patients on combination therapy, and who were non-compliant or discontinued the treatment anytime during the past 3 months were excluded. All patients received MTX monotherapy at a dose of 12.5–25 mg/week.

### 2.2. Clinical Assessment

A full medical history and a medical examination were conducted for all participants. Patient demographics, and clinicopathological and treatment characteristics were collected from clinical records. Data relevant to disease state and clinical response were recorded. These include age, gender, body-mass index, and smoking history. Regarding smoking, patients were classified either as current smokers, past smokers, or non-smokers. MTX clinical response was recorded and assessed using the DAS-28. DAS stands for “disease activity score”, and 28 refers to the number of joints clinically examined for this assessment. We calculated the DAS-28 using tender-joint count, swollen-joint count, and CRP; then, the patient global assessment was fed into the mathematical formula to produce the overall disease activity score. Adherence to MTX therapy was assessed via self-reporting during the monthly patient visits. A patient who missed two consecutive doses or three or more non-consecutive doses during an 8-week period was considered non-adherent and excluded from the study. According to response to MTX, patients were classified as responders or non-responders using the European League against Rheumatism (EULAR) response criteria based on DAS-28 scores, which was reported by Prevoo et al. [24]. All patients were evaluated for response at 3 and 6 months after initiation of MTX monotherapy. Two DAS-28 calculations were performed for each patient. Responders were patients who achieved either remission (DAS < 2.6) or low disease activity (DAS < 3.2) according to DAS 28 at the end point (6 months from initiating MTX therapy). Non-responders were patients who failed to accomplish either remission or low activity of disease on either of the two evaluations. Therefore, non-response to MTX was defined only after a minimum period of at least 6 months of MTX therapy. Response to MTX was defined when patients presented a DAS28 ≤ 3.2. In addition, blood samples were collected from all patients for calculating complete blood count (CBC), erythrocyte sedimentation rate (ESR), C-reactive protein (CRP), liver enzymes, serum creatinine, Rheumatoid Factor IgG (RF-IgG), and cyclic citrullinated peptide antibodies (anti-CCP), and conducting molecular analysis.

### 2.3. Molecular Analysis

About three milliliters of venous blood was collected in an EDTA anticoagulant vacutainer, and the collected blood was kept at −20 °C until DNA extraction.

#### 2.3.1. DNA Extraction

Invitrogen GeneCatcher purification system (Thermo Fisher, Waltham, MA, USA) was used to extract genomic DNA from the collected frozen venous blood according to the manufacturer’s instructions. NanoDrop spectrophotometer was used to determine the concentration and purity of DNA (NanoDrop Technologies, Inc., Wilmington, DE, USA).

#### 2.3.2. SNP Selection and Genotyping via Real-Time PCR

Ten genetic polymorphisms spanning four main genes in the cellular methotrexate pathway, namely *ATIC*, *SLC19A1*, *GGH*, and *DHFR*, were selected based on their variant annotations information and the association between a variant and a drug phenotype that were obtained from the PharmGKB database [25] (accessed from: https://www.pharmgkb.org, accessed on 9 June 2022) (Table 1).

The ten SNPs were assessed via real-time PCR using the TaqMan SNP Genotyping Assays (Applied Biosystems, Foster City, CA, USA), as shown in Table 2. The 10 studied SNPs were: three SNPs in *ATIC* gene (rs4673990, rs3821353, and rs16853834), three SNPs in the *SLC19A1* gene (rs11702425, rs9977268, and rs7279445), one SNP within the *GGH* gene (rs12681874), and three SNPs in the *DHFR* gene (rs12517451, rs10072026, and rs1643657). Applied Biosystems Step one plus Real-Time PCR detection system was used to conduct genotyping reactions and allelic discrimination, respectively. Each PCR mixture contained 12.5 µL TaqMan Universal master mixtures and primer with its specific primers, and probe 6-carboxyfluorescein [FAM] and [VIC] dyes were used to optimize the reaction. Each assay was performed under standard conditions as follows: inactivation at 50 °C for 2 min, then hold at 95 °C for 10 min, followed by 40 cycles of denaturation at 95 °C for 15 s and annealing at 60 °C for 1 min.

### 2.4. Statistical Analysis

Data were analyzed using Statistical Package for the Social Sciences (SPSS version 22.0, Chicago, IL, USA) software for windows. Demographic data were statistically presented using frequencies and percentages. Means and standard deviations were used to present parametric data, while medians and interquartile ranges were used to present the non-parametric data. For associations, the following tests were used to test differences for significance; chi-square, one-sample *t*-test, and one-way ANOVA with least significance difference. *p*-value was set at <0.05 for significant results. Analyses of allele frequencies and carriage rates were carried out. Genotype frequencies were assessed for deviation from the Hardy–Weinberg equation using an online program (http://www.oege.org/software/hwe-mr-calc.shtml, accessed on 10 January 2022). The relationship between allele frequencies and the response to MTX in patients with RA was determined under different genetic association models using odds ratio with multiple logistic regression analysis after adjustment of RA risk factors. 

### 2.5. Ethical Considerations

The approval for the study protocol was issued by the local Ethics Committee of the SCU Faculty of Medicine (Reference 4209). Informed consent was obtained from all participants according to the standards of the Helsinki Declaration.

## 3. Results

### 3.1. The Study Population Baseline Characteristics

This study included 107 Egyptian RA patients, 94 (88%) females and 13 (12%) males, with a mean age of 44.4 ± 11.4 years old, of which 28 (26%) were smokers. Considering the disease-related variables shown in Table 3, the mean disease duration was 6.27 ± 5.5 years, and the mean disease activity score (DAS28) was 4.7 ± 1.5. All the patients were on MTX monotherapy for a duration of 3.7 ± 3.3 years. Regarding laboratory features, the mean serum creatinine was 0.73 ± 0.31mg/dL, while the median (IQR) of CRP, RF, and anti-CCP in RA patients was 10 (6–200), 20 (18–64) and 30 (25–55), respectively. Non-response to MTX was observed in 40 (37.3%) patients. A comparison between MTX responders and non-responders is shown in Table 2. MTX non-responders had a longer duration of RA (*p* < 0.01), more tender and swollen joints (*p* < 0.01 each), higher VAS and DAS28 scores (*p* < 0.01 each), and higher ESR and CRP (*p* < 0.01 each).

### 3.2. Allelic Discrimination Analysis

Seven polymorphisms were in accordance with Hardy–Weinberg equilibrium (*ATIC* rs3821353: *p* = 0.15; *ATIC* rs4673990: *p* = 0.69; *ATIC* rs16853834: *p* = 0.79; *DHFR* rs12517451: *p* = 0.25; *SLC19A1* rs7279445: *p* > 0.999; *SLC19A1* rs11702425: *p* = 0.17; and *DHFR* rs10072026: *p* = 0.082), and three were in linkage disequilibrium (LD) (*SLC19A1* rs9977268: *p* = 0.0001; *DHFR* rs1643657: *p* = 0.0018; and *GGH* rs12681874: *p* = 0.0083).

On comparing the frequency of genotypes among the study groups, the three *ATIC* gene rs3821353 genotypes and the rs4673990 AA and AG genotypes were statistically significant between RA patients and the controls, with a *p*-value < 0.001 for both polymorphisms. Additionally, the *SLC19A1* gene rs11702425 CC and TT genotypes were statistically significant between RA patients and the controls, with a *p*-value = 0.005. The only polymorphism studied for the *GGH* gene, which was rs12681874 genotypes CC and CT, showed a statistically significant relationship between RA patients and the controls, with a *p*-value = 0.009. Finally, the three polymorphisms for the *DHFR* gene did not manifest any statistically significant differences between RA patients and the controls (Figure 1).

Regarding the comparison of variants between the study groups, the *ATIC* gene rs3821353 G variant was more frequent among RA patients (G allele: 72% in RA patients versus 57% in the control group, *p* = 0.001), and for *ATIC* gene rs4673990, variant G was more frequent among RA patients (G allele: 39% in RA patients versus 24% in the control group, *p* < 0.001). Similarly, a higher proportion of *SLC19A1* gene rs11702425 C variant was observed in RA patients (C allele: 48% in RA patients versus 32% in the control group, *p* < 0.001), while in *GGH* rs12681874, the T variant was more frequent in the normal control group (T allele: 14% in RA patients versus 26% in the control group, *p* = 0.002) (Figure 2).

For RA patients, the overall minor allele frequencies of *ATIC* genes rs3821353, rs4673990, and rs16853834 were 0.28 (T), 0.39 (G), and 0.24 (T), respectively. For *DHFR* genes rs12517451, rs10072026, and rs1643657, the overall minor allele frequencies were 0.21 (T), 0.17 (C), and 0.45 (A), respectively. For *SLC19A1* genes rs7279445, rs11702425, and rs9977268, and *GGH* gene rs12681874, the overall minor allele frequencies were 0.29 (C), 0.48 (C), 0.35 (C), and 0.14 (T), respectively. A comparison with other ethnic populations, from the 1000 Genome Project, is shown in Figure 3.

### 3.3. Association of Methotrexate Pathway Genes Variants with Disease Risk

Out of the 10 studied SNPs, 4 SNPs were associated with RA in our study.

#### 3.3.1. ATIC Gene rs3821353 

Carriers of allele *ATIC rs3821353* G allele were nearly 2 times more likely to develop RA under allelic comparison (OR = 1.94, 95% CI = 1.3 to 2.9, *p* = 0.001). However, *ATIC* rs3821353 genotype T/T was 0.28 times more likely to protect against RA under codominant comparison (OR = 0.28, 95% CI = 0.13−0.60, *p* = 0.001) and the recessive model (OR = 0.26, 95% CI = 0.12 to 0.54, *p* < 0.001) (Table 4).

#### 3.3.2. ATIC Gene rs4673990 

Carriers of the *ATIC* gene rs4673990 G allele were 2 times more likely to develop RA under allelic comparison (OR = 2, 95% CI = 1.33 to 3.07, *p* < 0.001). Additionally, the *ATIC* rs4673990 A/G genotype was 4.5 times more likely to develop RA under codominant comparison (OR = 4.5, 95% CI = 2.35 to 8.7, *p* < 0.001) and 2.2 times more likely to develop RA under over-dominant comparison (OR = 2.24, 95% CI = 1.22–4.11, *p* = 0.008), while the G/G genotype was 0.34 times more likely to protect against RA under recessive comparison (OR = 0.34, 95% CI = 0.16 to 0.7, *p* = 0.004) (Table 4).

#### 3.3.3. SLC19A1 Gene rs11702425

With respect to allele T for this polymorphism, it was 0.52 times more likely to protect against RA under allelic comparison (OR = 0.52, 95% CI = 0.35 to 0.77, *p* = 0.001). *SLC19A1* gene rs11702425 genotypes C/T and T/T were 0.38 and 0.26 times more likely to protect against RA under codominant comparison (OR = 0.38, 95% CI = 0.17 to 0.86, *p* = 0.02) (OR = 0.26, 95% CI = 0.11 to 0.6, *p* = 0.001), respectively. Additionally, they showed a 0.52-times greater protective effect with the T/T genotype under recessive comparison (OR = 0.52, 95% CI = 0.13 to 0.92, *p* = 0.02) (Table 3). Moreover, they also showed a 0.32-times greater protective effect with the C/T genotype under dominant comparison (OR = 0.32, 95% CI = 0.15 to 0.69, *p* = 0.003) (Table 4).

#### 3.3.4. SLC19A1 Gene rs7279445

Carriers of the SLC19A1 gene rs7279445 TT genotype were 1.84 times more likely to develop RA under recessive comparison (OR = 1.8, 95% CI = 1.06 to 3.18, *p* = 0.02).

#### 3.3.5. GGH Gene rs12681874 

Carrying the *GGH* gene rs12681874 T allele was 0.48 times more likely to protect against RA under allelic comparison (OR = 0.48, 95% CI = 0.3 to 0.78, *p* = 0.003). Additionally, *GGH* rs12681874 genotype C/T was 0.4 times more likely to protect against RA under codominant and over-dominant comparison (OR = 0.4, 95% CI = 0.2 to 0.75, *p* = 0.004 and OR = 0.42, 95% CI = 0.22 to 0.8, *p* = 0.008, respectively). Additionally, the C/T genotype showed a 0.4 times greater protective effect with the T/T genotype under dominant comparison (OR = 0.4, 95% CI = 0.22 to 0.73, *p* = 0.002) (Table 4).

### 3.4. Association of MTX Pathway Gene Variants and the Treatment Response to MTX

The genotype frequencies of the studied genetic variants in relation to MTX treatment response are shown in Figure 4. The overall association between genotype frequencies and MTX response revealed an association with only *ATIC* gene rs3821353 (*p* = 0.029); however, further analysis using genetic association models revealed that 3 out of 10 of the studied SNPs were related to MTX clinical response (Table 5).

#### 3.4.1. ATIC Gene rs3821353 and SLC19A1 Gene rs7279445

The *ATIC* rs3821353 G/T genotype was significantly associated with non-response to MTX treatment under codominant comparison, with a 0.33-fold chance for a positive response to MTX (OR = 0.33, 95% CI = 0.12−0.9, *p*-value = 0.034), indicating its bad prognostic significance for MTX treatment in RA patients. Additionally, both the C/T and C/C genotypes for *SLC19A1* gene rs7279445 were significantly associated with a negative response to MTX treatment under dominant comparison (OR = 5.86, 95% CI = 1.92−17.84, *p*-value = 0.0018) (Table 5).

#### 3.4.2. DHFR Gene rs10072026

On the other hand, both the C/T and C/C genotypes for *DHFR* gene rs10072026 were significantly associated with a positive response to MTX treatment under dominant comparison, with a 0.05-fold change for a negative response to MTX (OR = 0.05, 95% CI = 0.02−0.17, *p*-value = 0.0001), indicating their good prognostic significance for MTX treatment in RA patients (Table 5).

### 3.5. Haplotypes of Methotrexate Pathway Gene Variants and the Clinical Response to MTX

Table 6 represents the relationship between MTX pathway gene haplotypes and the clinical response to MTX. Our results did not show statistically significant relationships among all variants under study and the response to MTX therapy. The “TTG” haplotype of the DHFR three gene variants with an odds ratio equivalent to 1.76, the “TAT” haplotype of the *ATIC* three gene variants with an odds ratio equivalent to 3.11, and the “CCC” haplotype of the *SLC19A1* three gene variants with an odds ratio equivalent to 2.80 were associated with higher odds for non-response to MTX therapy compared to all other haplotypes. Therefore, our results pinpoint that the haplotypes studied are associated with a high possibility of non-response to treatment, given that the data were not statistically significant.

## 4. Discussion

Methotrexate (MTX) is the first-line drug for the treatment of RA. However, in about 30% of cases, MTX is insufficient as a monotherapy. Moreover, a further 30% of patients suffer from severe adverse effects. The potential to personalize MTX treatment to meet patients’ exact needs is an essential target. It would be beneficial if it was clinically applied. Therefore, pharmacogenomic approaches are rapidly becoming popular for discovering biomarkers of disease remission and toxicity. The present work aims to evaluate selected SNPs in genes encoding for proteins involved in MTX pathways as potential predictors of MTX therapeutic outcomes among Egyptian RA patients.

Our study identified 10 SNPs in four master MTX metabolic pathway genes, showing evidence for an association with the potency of MTX in a cohort of Egyptian RA patients. Several studies were performed to investigate MTX response in RA patients, mostly relying on a target-gene approach, genotyping single SNPs in each gene, and assessing the association. The current study targeted MTX metabolic pathway candidate genes owing to the previous success of this approach for various types of treatments (i.e., detecting the response of vitamin K and cytochrome p450 gene polymorphisms to warfarin) [26,27,28,29]. 

Four of the studied SNPs were protective against RA—namely, *ATIC* rs3821353 (in its codominant and recessive models), *ATIC* rs4673990 (in its recessive model), *SLC19A1* rs11702425 (in its dominant C/T and codominant T/T models), and *GGH* rs12681874 (in its dominant, codominant, and over-dominant models)—while only two SNPs were predictive of increase in RA risk: *ATIC* rs3821353 (in its over-dominant model), and *ATIC* rs4673990 (in its codominant and over-dominant models). Interestingly, *ATIC* rs4673990 was more likely to protect against RA under recessive comparison G/G. Regarding allelic variants, carriers of the *ATIC* rs3821353 G allele and *ATIC* rs4673990 G allele were more susceptible to RA. While carriers of the *SLC19A1* rs11702425 T allele and *GGH* rs12681874 T allele were protected from RA.

Non-response to MTX was associated with *ATIC* rs3821353 in its codominant G/T model, as well as with *SLC19A1* rs7279445 in its dominant forms’ C/T and C/C, indicating their bad prognostic significance for MTX treatment in RA patients. Meanwhile, MTX’s good response was associated with *DHFR* rs10072026 in its dominant forms’ C/T and C/C.

Regarding the functional effect of these SNPs, with a significant association with MTX treatment response in the present study, the published literature is mainly reporting their role in the MTX pathway and folate metabolism. However, the precise biological implication of each SNP alteration in each gene is not clear yet. However, in silico data suggest that these SNPs may have a direct role in regulating the expression of their target genes. In this regard, SNPs in the ATIC gene, including rs3821353, can affect the bifunctional protein that catalyzes the last two steps of the de novo purine biosynthetic pathway [30]. Additionally, it can cause alterations in AMP-activated protein kinase (AMPK) signaling pathways [30]. For the DHFR gene, SNPs within this gene, including rs10072026, can affect the metabolism of water-soluble vitamins and cofactors, fluoropyrimidine activity, cell cycle, and mitotic G1 phase and G1/S transition pathways [31]. Finally, the SLC19A1 gene SNPs, including rs7279445, can affect the metabolism of water-soluble vitamins and cofactors, metabolism, and alterations in histone modifications pathway [32].

Considering de novo purine synthesis, we studied three SNPs in the *ATIC* gene: rs4673990, rs3821353, and rs16853834. Only the rs3821353 G/T genotype was associated with non-response to MTX in our study. Previously, Owen et al. confirmed a significant association between rs16853834 and clinical response to MTX, but not with MTX-related toxicity [14]. Additionally, the T allele for rs12995526 was associated with a better response [33]. 

In prior RA studies, it was documented that SNP (rs2372536) located in exon 5 of the *ATIC* gene is associated with response to MTX [20,34,35,36]. On the other hand, other studies did not find such an association [14,36,37]. The variations seen in the current study and other studies are probably explained by small sample sizes; inter-study variability; different outcomes to measure efficacy; and comparing patients at various stages of disease. Additionally, a number of SNPs in the *ATIC* gene were related to the response to MTX, such as rs12995526 [14]. Different research results document that SNPs in the same gene show evidence for an association with the response to MTX, suggesting that this gene might be involved in therapeutic efficacy outcome. Several variants within the gene may contribute.

For the polyglutamation pathway, our results showed that neither the genotypes nor the alleles of the *GGH* gene (rs12681874) were associated with MTX therapeutic outcomes, as reported previously [4,20,33,34]. However, in another study, the rs12681874 C allele was associated with MTX non-response in RA [33]. 

MTX enters the cells via the *SLC* gene family and it is transported out via the *ABC* family of transporters [38,39,40]. The main member of the *SLC* family is solute carrier family 19 member 1 (*SLC19A1*), which is also known as reduced folate carrier 1 (*RFC1*) [41]. Genetic polymorphisms in *RFC1* are thought to influence *RFC1* function and affect MTX transport [42,43]. However, we did not find a relationship between the three studied SNPs of the *SLC19A1* gene and clinical response to MTX in our study. Several previous studies have assessed the *SLC19A1* gene’s role, focusing on SNP (rs1051266), which results in substituting arginine with histidine at codon 27 in the first transmembrane domain of the RFC protein [44,45,46,47,48,49]. Many reports have suggested an association between this SNP and MTX efficacy [44,45,47,48]. However, Owen and co-workers did not find such an association; instead, they found another six SNPs in the *SLC19A1* gene and its neighboring gene, *COL18A1*, that were associated with MTX efficacy. Owing to the high polymorphic nature of the *SLC19A1* gene in humans and the strong LD across the gene, some of these variants were correlated with other polymorphisms that were less commonly reported in the RA literature [34]. This suggests that the overall contribution of polymorphisms cannot be explained by the most-reported SNP (rs1051266) alone, and that other variant within the gene may also be crucial in influencing the response to MTX. 

To assess the clinical response variability among different patient groups in RA patients under MTX treatment, a study involving twenty-three SNPs in *SLC* and *ABC* MTX transporters evaluated the influence of these SNPs as important predictors to MTX in Portuguese RA patients. *SLC22A11* rs11231809 T carrier multivariate analyses demonstrated a significant association with non-response to MTX, while *SLC19A1* variants, which are commonly studied SNPs, were not significant for clinical response in their study. This could be due to other polymorphisms’ existence in these transporters, which could ameliorate their function or the other transporters’ expression in target cells, thus, equalizing the influx/efflux ratio [4]. 

Regarding the relationship between MTX pathway gene haplotypes and the clinical response to MTX, our results did not show statistically significant relationships among all study variants and the response to MTX therapy. However, we found that the “TTG” haplotype of the three *DHFR* gene variants with an odds ratio equivalent to 1.76, the “TAT” haplotype of the three *ATIC* gene variants with an odds ratio equivalent to 3.11, and the “CCC” haplotype of the three *SLC19A1* gene variants with an odds ratio equivalent to 2.80 were associated with higher odds for non-response to MTX therapy compared to all other haplotypes.

Due to lack of studies that analyze the impact of the studied SNPs in protein function and/or MTX therapeutic outcome, further evidence is essential to support the interpretation of our results. Moreover, a larger cohort with precise clinical measures and a well-defined outcome is required to confirm these SNP association with MTX treatment response. Once confirmed, the causal variants can be determined via fine mapping to provide more biological insights into the mechanisms by which MTX response is determined.

All our patients were enrolled within the same geographical area, with homogenous ethnicity, which is a point of strength in our study. However, our study has some limitations: Firstly, as we lack complete knowledge about the MTX metabolic pathway, we may have failed to investigate other important genes. Additionally, although we reported several SNPs associated with therapeutic efficacy in this study, we suggest that combinations of risk and SNP response to MTX will be more predictive than individual SNP effects, as shown previously [34,36,50]. Secondly, the study is monocenter and its sample size is limited (n = 107); nevertheless, this represents one of the biggest Egyptian studies to date in the context of MTX pharmacogenomics in RA. Thirdly, data collection in retrospective studies has known biases such as missing information related to important clinical variables that might be predictive of MTX therapeutic response.

In summary, our results confirm some of the previous literature findings in RA. At the same time, we reported various associations between the *GGH*, *ATIC*, *DHFR*, and *SLC19A1* (*RFC*) MTX metabolic pathway genes and both disease risk and the efficacy of MTX therapeutic response. Finally, we provided more evidence supporting the *ATIC* gene’s role in MTX therapeutic response. Several SNPs under study are located in non-coding regions; therefore, we cannot predict their impact on the functional properties of the gene and, therefore, it is not possible to propose the specific biological pathways involved.

## Figures and Tables

**Figure 1 diagnostics-12-01560-f001:**
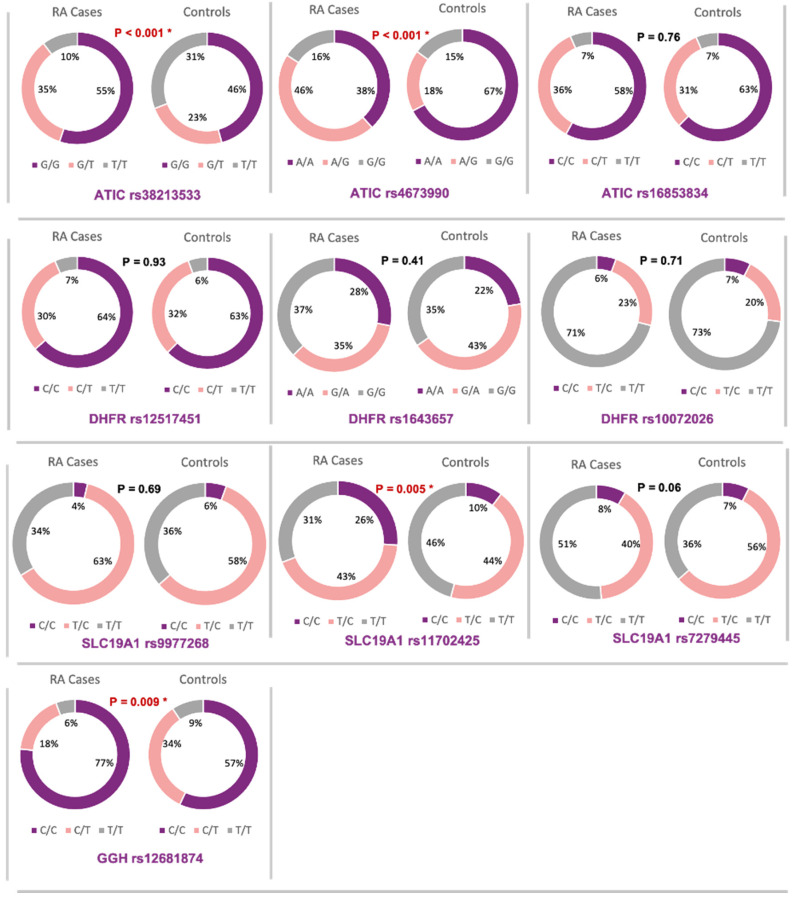
Genotype frequencies of methotrexate pathway genes in rheumatoid arthritis versus normal controls. * *p*-value < 0.05 and statistically significant.

**Figure 2 diagnostics-12-01560-f002:**
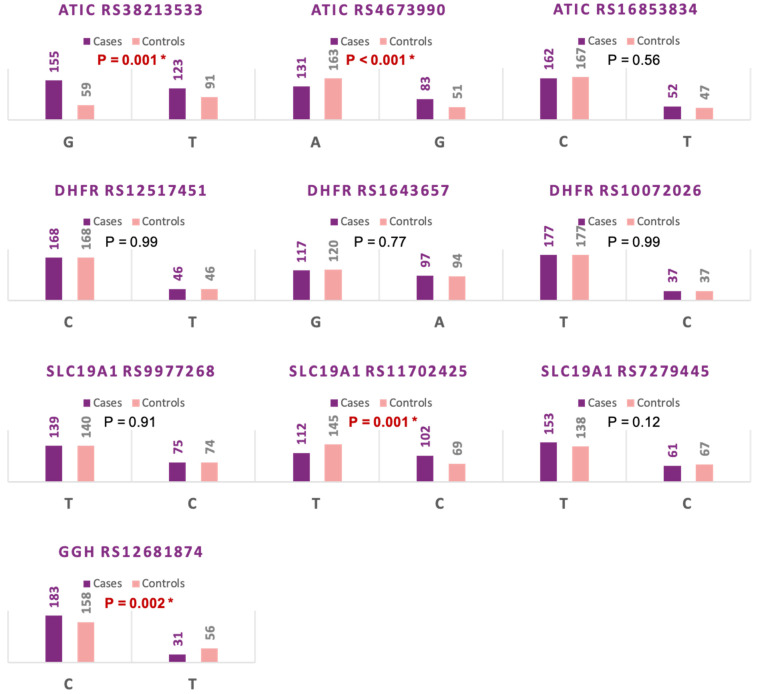
Allele frequencies of methotrexate pathway genes in rheumatoid arthritis versus normal controls. * *p*-value < 0.05 and statistically significant.

**Figure 3 diagnostics-12-01560-f003:**
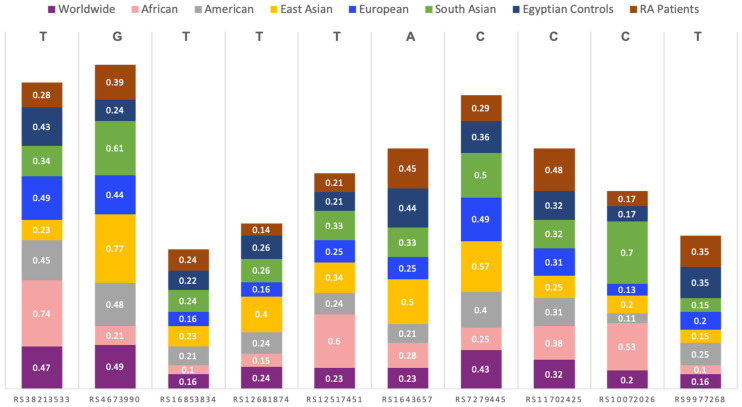
Comparison between the study groups’ minor allele frequencies and those of other ethnic populations from 1000 Genome Project (https://www.ensembl.org/) accessed on 20 January 2022.

**Figure 4 diagnostics-12-01560-f004:**
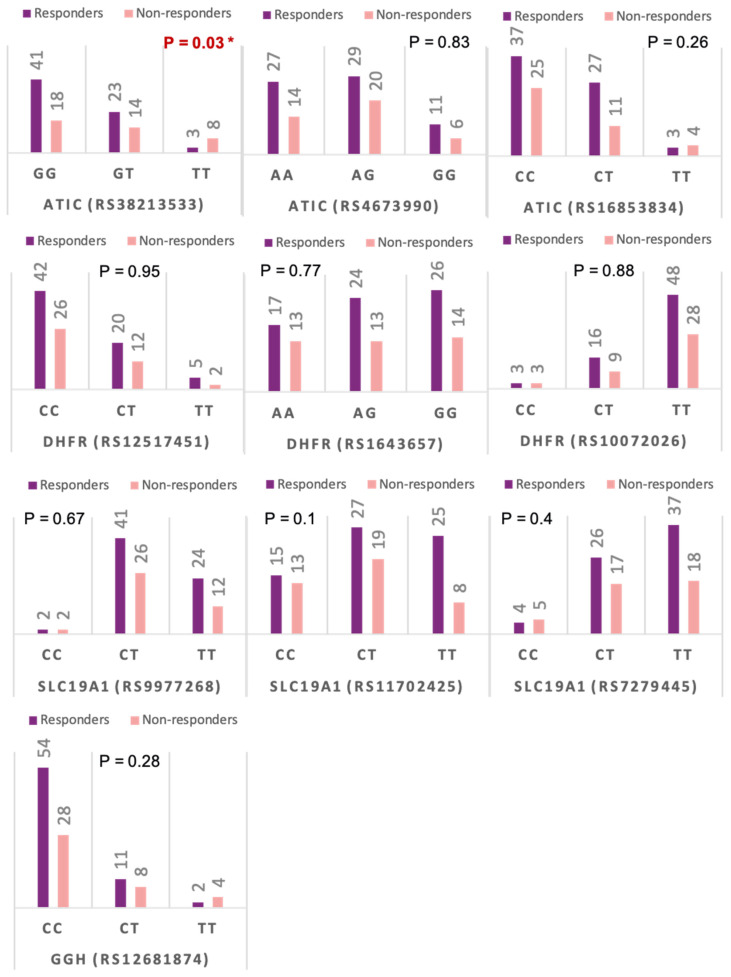
Genotype frequencies of the studied genetic variants in relation to methotrexate treatment response. * *p*-value < 0.05 and statistically significant.

**Table 1 diagnostics-12-01560-t001:** Characteristics and consequences of the ten SNPs understudy in MTX pathway and their drug response data from PharmGKB.

Gene	SNP	Genomic Location	Alleles	RA Response and Toxicity (Association Information)
**ATIC**	rs4673990	Introns	A > G	Allele A is linked to lesser response with MTX
	rs16853834	Introns	C > T	Allele T is linked with better MTX therapy response than allele G
	rs3821353	Introns	G > T	Failure to respond to MTX is associated with allele T
**SLC19A1**	rs11702425	Synonymous	T > A T > C T > G	Allele C is associated with lesser MTX treatment response
	rs9977268	Introns	C > T	Allele T is associated with lesser MTX treatment response
	rs7279445	Introns	C > T	MTX resistance has been linked to the presence of allele T
**GGH**	rs12681874	Introns	C > T	RA patients with allele C show lesser MTX treatment response
**DHFR**	rs12517451	5′ Flanking region	C > A	RA patients are more likely to experience side effects from MTX if they have allele T
			C > G	
			C > T	
	rs10072026	Introns	T > C	RA patients are less likely to experience MTX side effects if they have allele C
	rs1643657	Introns	T > C	RA patients with allele C are less likely to experience MTX side effects

**Table 2 diagnostics-12-01560-t002:** TaqMan SNP genotyping assay details included in the study.

Gene	SNP rs#	SNP ID	Context Sequence [VIC/FAM]
**ATIC**	rs4673990	C_28017839_30	TAGATTCCATGGTACCATGTTGAGA**[A/G]**TTGTGCCTAGCTACTGAGAGTCTTT
	rs16853834	C_33295734_10	ATATTACACTTCTTTCCTAGTGTCC**[C/T]**GAGCCTCAGAATACAAGATGGAGCT
	rs3821353	C_362253_20	TATCAAAGTGATATCAAGCAGAACA**[G/T]**AGAGAAAGAGTGAGTCAGTAATGAA
**SLC19A1**	rs11702425	C_2176981_10	AGGGACCTCCCGGCCTGCCGGGACT**[C/T]**AAGGTCAGTGACGGATATGTCTGGG
	rs9977268	C_25766978_10	AAGCATGTCCCACCCTCCTCTCGGG**[C/T]**AGTGCCACCCCAGGGAGGGGTCCTT
	rs7279445	C_31153754_20	AAGCATGTCCCACCCTCCTCTCGGG**[C/T]**AGTGCCACCCCAGGGAGGGGTCCTT
**GGH**	rs12681874	C_11852315_10	TTAAGAGTAATGGTGAATATTTTTT**[C/T]**CCAATTTACCTGAAAAAAAAAATCA
**DHFR**	rs12517451	C_32167960_10	GCTTTATTCCCCTTTATCCCTGTGA**[C/T]**GGCGGGGGCCTGTAATAATATCTTG
	rs10072026	C_3103314_10	AATAGCTCCTTTTATACAATTTCAT**[C/T]**TTATCATATACTGATCTCCACTATG
	rs1643657	C_3103233_10	TAGTCACATATTTCACTGCTGAATT**[C/T]**CTTTCCTATAATTATTTTAAACCAA

**Table 3 diagnostics-12-01560-t003:** The clinical and laboratory features of rheumatoid arthritis patients’ responders vs. non-responders to MTX.

Variable	Responders (n = 67) mean ± SD	Non-Responders (n = 40) mean ± SD	*p*-Value
RA duration (Yrs.)	4.75 ± 4.12	8.01 ± 6.56	**<0.01 ****
MTX duration (Yrs.)	3.23 ± 2.33	4.34 ± 4.26	0.11
Current smoking—no (%) Past smoking—no (%) Non-smoking—no (%)	6 (9.0) 10 (14.9) 51 (76.1)	2 (5.0) 10 (25.0) 28 (70.0)	0.37 †
Tender-joint count	4.65 ± 3.33	14.67 ± 8.48	**<0.01 ****
Swollen-joint count	0.35 ± 0.81	4.51 ± 4.35	**<0.01 ****
VAS for pain (cm)	5.67 ± 1.89	8.71 ± 1.36	**<0.01 ****
DAS-28	2.50 ± 0.53	5.59 ± 1.17	**<0.01 ****
Hemoglobin level (g/dL)	11.94 ± 1.5	11.44 ± 1.3	0.11
Platelet count (×10/mm^3^)	278.23 ± 62.5	288.23 ± 80.5	0.53
Total leucocytic count (×10/mm^3^)	8.21 ± 2.33	7.77 ± 2.35	0.39
ALT (IU/L)	19.51 ± 17.1	19.03 ± 8.7	0.87
AST (IU/L)	20.02 ± 9.7	19.28 ± 7.1	0.69
Serum creatinine (mg/dL)	0.78 ± 0.4	0.67 ± 0.1	0.10
ESR (mm/hr)	45.19 ± 24.4	59.19 ± 24.2	**<0.01 ****
CRP (mg/dL)	9.64 ± 6.50	27.21 ± 19.10	**<0.01 ****
Rheumatoid factor (median, (IQR))	20 (11–32)	20 (20–67)	0.96
Anti-CCP (median, (IQR))	30 (21.25–64.13)	30 (25–40)	0.78

** *p*-value significant at <0.01, † *p*-value calculated using Chi Square test. MTX—methotrexate; VAS—Visual Analogue Scale; DAS28—disease activity score 28; ESR—erythrocyte sedimentation rate; CRP—C-reactive protein; AST—aspartate transaminase; ALT—alanine transaminase; anti-CCP—anti-citrullinated protein antibody; RA—rheumatoid arthritis. RF and anti CCP were analyzed using Mann–Whitney U test for non-parametric variables.

**Table 4 diagnostics-12-01560-t004:** Genetic association models for methotrexate pathway genes with rheumatoid arthritis disease risk.

SNP	Model	Genotype	RA Patients	Controls	Odds Ratio (95% CI)	*p*-Value
***ATIC*** **Gene** **rs3821353**	Codominant	G/G	59	49	Reference	
		G/T	37	25	1.23 (0.65–2.32)	0.52
		T/T	11	33	0.28 (0.13–0.60)	**0.001 ***
	Dominant	G/G	59	49	Reference	
		G/T-T/T	48	58	0.69 (0.40–1.18)	0.17
	Recessive	G/G-G/T	96	74	Reference	
		T/T	11	33	0.26 (0.12–0.54)	**0.000 ***
	Over-dominant	G/G-T/T	70	82	Reference	
		G/T	37	25	1.7 (0.95–3.16)	0.07
	Allelic Model	T	59	91	Reference	
		G	155	123	1.94 (1.3–2.9)	**0.001 ***
***ATIC*** **Gene** **rs4673990**	Codominant	A/A	41	72	Reference	
		A/G	49	19	4.5 (2.35–8.7)	**0.000 ***
		G/G	17	16	1.86 (0.85–4.08)	0.12
	Dominant	A/A	59	49	Reference	
		A/G-G/G	66	35	1.56 (0.9–2.7)	0.11
	Recessive	A/A-A/G	90	91	Reference	
		G/G	11	33	0.34 (0.16–0.7)	**0.004 ***
	Over-dominant	A/A-G/G	58	88	Reference	
		A/G	37	25	2.24 (1.22–4.11)	**0.008 ***
	Allelic Model	A	131	163	Reference	
		G	83	51	2 (1.33–3.07)	**0.000 ***
***ATIC*** **Gene** **rs16853834**	Codominant	C/C	62	67	Reference	
		C/T	38	33	1.24 (0.69–2.22)	0.46
		T/T	7	7	1.08 (0.36–3.25)	0.89
	Dominant	C/C	62	67	Reference	
		C/T-T/T	45	40	1.2 (0.7–2.1)	0.48
	Recessive	C/C-C/T	100	100	Reference	
		T/T	7	7	1.00 (0.3–2.9)	1.00
	Over-dominant	C/C-T/T	69	74	Reference	
		C/T	38	33	1.23 (0.7–2.1)	0.47
	Allelic Model	C	162	167	Reference	
		T	52	47	1.14 (0.73–1.79)	0.56
***DHFR*** **Gene** **rs12517451**	Codominant	C/C	68	67	Reference	
		C/T	32	34	0.92 (0.5–1.67)	0.8
		T/T	7	6	1.15 (0.37–3.6)	0.81
	Dominant	C/C	68	67	Reference	
		C/T-T/T	39	40	0.96 (0.55–1.67)	0.88
	Recessive	C/C-C/T	100	101	Reference	
		T/T	7	6	1.18 (0.38–3.63)	0.77
	Over-dominant	C/C-T/T	75	73	Reference	
		C/T	32	34	0.9 (0.51–1.63)	0.76
	Allelic Model	C	168	168	Reference	
		T	46	46	1.00 (0.63–1.6)	1.0
***DHFR*** **Gene** **rs1643657**	Codominant	A/A	30	24	Reference	
		A/G	37	46	0.64 (0.32–1.28)	0.21
		G/G	40	37	0.86 (0.43–1.73)	0.68
	Dominant	A/A	30	24	Reference	
		A/G-G/G	77	83	0.74 (0.4–1.38)	0.34
	Recessive	A/A-A/G	67	70	Reference	
		G/G	40	37	1.12 (0.64–1.97)	0.67
	Over-dominant	A/A-G/G	70	61	Reference	
		A/G	37	46	0.92 (0.53–1.6)	0.77
	Allelic Model	A	97	94	Reference	
		G	117	120	0.94 (0.64–1.38)	0.77
***DHFR*** **Gene** **rs10072026**	Codominant	C/C	6	8	Reference	
		C/T	25	21	1.58 (0.47–5.3)	0.45
		T/T	76	78	1.3 (0.43–3.9)	0.64
	Dominant	C/C	6	8	Reference	
		C/T-T/T	101	99	1.36 (0.45–4.06)	0.58
	Recessive	C/C-C/T	31	29	Reference	
		T/T	76	78	0.91 (0.50–1.65)	0.76
	Over-dominant	C/C-T/T	82	86	Reference	
		C/T	25	21	1.25 (0.65–2.4)	0.5
	Allelic Model	C	37	37	Reference	
		T	177	177	1.00 (0.6–1.6)	1.00
***SLC19A1*** **Gene** **rs9977268**	Codominant	C/C	4	6	Reference	
		C/T	67	62	1.62 (0.43–6.01)	0.47
		T/T	36	39	1.38 (0.36–5.3)	0.63
	Dominant	C/C	4	6	Reference	
		C/T-T/T	103	101	1.52 (0.42–5.58)	0.52
	Recessive	C/C-C/T	71	68	Reference	
		T/T	36	39	0.88 (0.5–1.5)	0.66
	Over-dominant	C/C-T/T	40	45	Reference	
		C/T	67	62	1.2 (0.7– 2.1)	0.48
	Allelic Model	C	75	74	Reference	
		T	139	140	0.98 (0.68–1.46)	0.91
***SLC19A1*** **Gene** **rs11702425**	Codominant	C/C	28	11	Reference	
		C/T	46	47	0.38 (0.17–0.86)	**0.02 ***
		T/T	33	49	0.26 (0.11–0.6)	**0.001 ***
	Dominant	C/C	28	11	Reference	
		C/T-T/T	79	96	0.32 (0.15–0.69)	**0.003 ***
	Recessive	C/C-C/T	74	58	Reference	
		T/T	33	49	0.52 (0.3– 0.92)	**0.02 ***
	Over-dominant	C/C-T/T	61	60	Reference	
		C/T	46	47	0.96 (0.56–1.65)	0.89
	Allelic Model	C	102	69	Reference	
		T	112	145	0.52 (0.35–0.77)	**0.001 ***
***SLC19A1*** **Gene** **rs7279445**	Codominant	C/C	9	8	Reference	
		C/T	43	60	0.63 (0.22–1.78)	0.39
		T/T	55	39	1.25 (0.44–3.53)	0.67
	Dominant	C/C	9	8	Reference	
		C/T-T/T	98	99	0.88 (0.32–2.37)	0.8
	Recessive	C/C-C/T	52	68	Reference	
		T/T	55	39	1.84 (1.06–3.18)	**0.02 ***
	Over-dominant	C/C-T/T	64	47	Reference	
		C/T	43	60	0.82 (0.49–1.38)	0.47
	Allelic Model	C	61	76	Reference	
		T	153	138	1.38 (0.91–2.07)	0.12
***GGH*** **Gene** **rs12681874**	Codominant	C/C	82	61	Reference	
		C/T	19	36	0.4 (0.2–0.75)	**0.004 ***
		T/T	6	10	0.45 (0.15–1.29)	0.14
	Dominant	C/C	82	61	Reference	
		C/T-T/T	25	46	0.4 (0.22–0.73)	**0.002 ***
	Recessive	C/C-C/T	101	97	Reference	
		T/T	6	10	0.57 (0.20–1.64)	0.3
	Over-dominant	C/C-T/T	88	71	Reference	
		C/T	19	36	0.42 (0.22–0.8)	**0.008 ***
	Allelic Model	C	183	158	Reference	
		T	31	56	0.48 (0.3–0.78)	**0.003 ***

* *p*-value < 0.05 and statistically significant. Abbreviations: CI, confidence interval; SNP, single nucleotide polymorphism.

**Table 5 diagnostics-12-01560-t005:** Genetic association models for methotrexate pathway genes with the treatment response to methotrexate.

SNP	Model	Genotype	MTX Respondent	MTX Non-Respondent	Odds Ratio (95% CI)	*p*-Value
***ATIC*** **Gene** **rs3821353**	Codominant	G/G	14	45	Reference	-
		G/T	7	30	0.33 (0.12–0.9)	**0.034 ***
		T/T	2	9	1.4 (0.27–7.2)	0.68
	Dominant	G/G	14	45	Reference	-
		G/T-T/T	9	39	1.35 (0.53–3.45)	0.62
	Recessive	G/G-G/T	21	75	Reference	-
		T/T	2	9	1.26 (0.25–6.28)	0.78
	Over-dominant	G/G-T/T	16	54	Reference	-
		G/T	7	30	1.2 (0.47–3.4)	0.64
	Allelic Model	T	48	11	Reference	-
		G	120	35	1.27 (0.6–2.7)	0.53
***ATIC*** **Gene** **rs4673990**	Codominant	A/A	10	31	Reference	-
		A/G	9	40	1.43 (0.52–3.96)	0.49
		G/G	4	13	1.05 (0.28–3.96)	0.94
	Dominant	A/A	10	31	Reference	-
		A/G-G/G	13	53	1.3 (0.5–3.35)	0.56
	Recessive	A/A-A/G	19	71	Reference	-
		G/G	4	13	0.9 (0.25–2.97)	0.82
	Over-dominant	A/A-G/G	14	44	Reference	-
		A/G	9	40	1.4 (0.55–3.62)	0.47
	Allelic Model	A	29	102	Reference	-
		G	17	66	1.1 (0.56–2.16)	0.77
***ATIC*** **Gene** **rs16853834**	Codominant	C/C	12	50	Reference	-
		C/T	9	29	0.77 (0.29–2.05)	0.6
		T/T	2	5	0.6 (0.1–3.48)	0.57
	Dominant	C/C	12	50	Reference	-
		C/T-T/T	11	34	0.74 (0.29–1.87)	0.53
	Recessive	C/C-C/T	21	79	Reference	-
		T/T	2	5	0.66 (0.12–3.67)	0.64
	Over-dominant	C/C-T/T	14	55	Reference	-
		C/T	9	29	0.82 (0.32–2.12)	0.68
	Allelic Model	C	33	129	Reference	-
		T	13	39	0.77 (0.37–1.6)	0.48
***DHFR*** **Gene** **rs12517451**	Codominant	C/C	14	54	Reference	-
		C/T	7	25	0.9 (0.3–2.6)	0.88
		T/T	2	5	0.6 (0.1–3.7)	0.62
	Dominant	C/C	14	54	Reference	-
		C/T-T/T	9	30	0.86 (0.3–2.23)	0.76
	Recessive	C/C-C/T	21	79	Reference	-
		T/T	2	5	0.66 (0.12–3.67)	0.63
	Over-dominant	C/C-T/T	16	59	Reference	-
		C/T	7	25	0.97 (0.35–2.64)	0.95
	Allelic Model	C	35	133	Reference	-
		T	11	35	0.83 (0.39–1.81)	0.65
***DHFR*** **Gene** **rs1643657**	Codominant	A/A	5	25	Reference	-
		A/G	10	27	0.54 (0.16–1.8)	0.3
		G/G	8	32	0.8 (0.23–2.74)	0.72
	Dominant	A/A	5	25	Reference	-
		A/G-G/G	18	59	0.65 (0.2–1.96)	0.45
	Recessive	A/A-A/G	15	52	Reference	-
		G/G	8	32	1.15 (0.44–3.02)	0.77
	Over-dominant	A/A-G/G	13	57	Reference	-
		A/G	10	27	0.61(0.24–1.58)	0.31
	Allelic Model	A	20	77	Reference	-
		G	26	91	0.9 (0.47–1.75)	0.78
***DHFR*** **Gene** **rs10072026**	Codominant	T/T	18	58	Reference	-
		C/T	5	20	1.24 (0.41–3.78)	0.7
		C/C	0	6	4.1 (0.22–76.55)	0.34
	Dominant	T/T	18	58	Reference	-
		C/T-C/C	26	5	0.05 (0.02–0.17)	**0.0001 ***
	Recessive	T/T-C/T	78	23	Reference	-
		C/C	6	0	0.25 (0.01–4.7)	0.36
	Over-dominant	C/C-T/T	18	64	Reference	-
		C/T	5	20	1.1 (0.4–3.4)	0.84
	Allelic Model	C	5	32	Reference	-
		T	41	136	0.5 (0.19–1.4)	0.2
***SLC19A1*** **Gene** **rs9977268**	Codominant	T/T	7	29	Reference	-
		C/T	15	52	0.84 (0.3–2.3)	0.72
		C/C	1	3	0.72 (0.06–8.05)	0.8
	Dominant	T/T	7	29	Reference	-
		C/T-C/C	16	55	1.23 (0.12–12.4)	0.86
	Recessive	T/T-C/T	22	81	Reference	-
		C/C	1	3	0.82 (0.30–2.25)	0.71
	Over-dominant	C/C-T/T	8	32	Reference	-
		C/T	15	52	0.87 (0.33–2.27)	0.77
	Allelic Model	C	17	58	Reference	-
		T	29	110	1.11 (0.56–2.19)	0.76
***SLC19A1*** **Gene** **rs11702425**	Codominant	C/C	5	23	Reference	-
		C/T	9	37	0.89 (2.66–3)	0.86
		T/T	9	24	0.58 (0.17–2)	0.39
	Dominant	C/C	5	23	Reference	-
		C/T-T/T	18	61	0.74 (0.25–2.22)	0.59
	Recessive	C/C-C/T	14	60	Reference	-
		T/T	9	24	0.62 (0.24–1.63)	0.33
	Over-dominant	C/C-T/T	14	47	Reference	-
		C/T	9	37	1.22 (0.48–3.14)	0.67
	Allelic Model	C	19	83	Reference	-
		T	27	85	0.72 (0.37–1.4)	0.33
***SLC19A1*** **Gene** **rs7279445**	Codominant	T/T	12	43	Reference	-
		C/T	10	33	0.92 (0.35–2.4)	0.87
		C/C	1	8	2.23 (0.25–19.65)	0.47
	Dominant	T/T	12	8	Reference	-
		C/T-C/C	11	43	5.86 (1.92–17.84)	**0.0018 ***
	Recessive	T/T-C/T	11	41	Reference	-
		C/C	12	43	0.96 (0.38–2.42)	0.93
	Over-dominant	C/C-T/T	13	51	Reference	-
		C/T	10	33	0.84 (0.33–2.14)	0.72
	Allelic Model	C	12	49	Reference	-
		T	34	119	0.86 (0.41–1.79)	0.68
***GGH*** **Gene** **rs12681874**	Codominant	C/C	16	66	Reference	-
		C/T	6	13	0.52 (0.17–1.6)	0.26
		T/T	1	5	1.2 (0.13–11.1)	0.86
	Dominant	C/C	66	16	Reference	-
		C/T-T/T	18	7	1.6 (0.57–4.5)	0.37
	Recessive	C/C-C/T	79	22	Reference	-
		T/T	5	1	0.7 (0.08–6.5)	0.77
	Over-dominant	C/C-T/T	71	17	Reference	-
		C/T	13	6	2.09 (0.7–6.3)	0.19
	Allelic Model	C	38	145	Reference	-
		T	8	23	0.75 (0.31–1.81)	0.53

* *p*-value < 0.05 and statistically significant. Abbreviations: CI, confidence interval; MTX, methotrexate; SNP, single nucleotide polymorphism.

**Table 6 diagnostics-12-01560-t006:** Haplotypes of methotrexate pathway gene variants and the treatment response to methotrexate.

Haplotype			Frequency	OR (95% CI)	*p*-Value
*DHFR* Gene Variants					
**(rs10072026)**	**(rs12517451)**	**(rs1643657)**			
T	C	A	0.327	Reference	-
T	C	G	0.3052	0.83 (0.36–1.92)	0.66
C	C	G	0.1183	0.87 (0.31–2.49)	0.8
T	T	G	0.1037	1.76 (0.64–4.85)	0.27
T	T	A	0.0912	0.75 (0.18–3.20)	0.7
***ATIC*** **Gene Variants**					
**(rs3821353)**	**(rs4673990)**	**(rs16853834)**			
G	A	C	0.3275	Reference	-
G	G	C	0.2387	0.89 (0.32–2.46)	0.82
G	A	T	0.1505	1.09 (0.35–3.37)	0.88
T	G	C	0.1044	1.09 (0.31–3.79)	0.89
T	A	C	0.0865	0.58 (0.12–2.89)	0.51
T	A	T	0.0477	3.11 (0.46–20.81)	0.24
***SLC19A1*** **Gene Variants**					
**(rs7279445)**	**(rs11702425)**	**(rs9977268)**			
T	C	T	0.2962	Reference	-
T	T	T	0.1851	1.53 (0.51–4.57)	0.45
T	T	C	0.1475	2.15 (0.57–8.10)	0.26
C	T	T	0.1333	2.28 (0.64–8.10)	0.21
T	C	C	0.0862	1.68 (0.30–9.51)	0.56
C	C	C	0.0593	2.80 (0.41–18.92)	0.29

Statistical significance was considered at *p*-value < 0.05. Abbreviations: CI, confidence interval; OR, odds ratio.

## Data Availability

The datasets used and/or analyzed during the current study are available from the corresponding author on reasonable request.

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
