# Peer review of "Pharmacogenomics of Methotrexate Pathway in Rheumatoid Arthritis Patients: Approach toward Personalized Medicine"

_diagnostics, 2022, doi:10.3390/diagnostics12071560_

Round 1

Reviewer 1 Report

Some observations.

1. Methodology: How was the adherence of the patients to the treatment evaluated during 3 months?

2. Line 115: Since it is the first time that the abbreviation DAS28 is used, it is suggested to add the meaning.

3. Line 117 and 123: CRP add meaning of abbreviation. Idem CBC, ESR, S., RF

4. Methodology, line 151: It is mentioned that means and standard deviation were used; however, the results (table 2) also show medians. Please specify in the methodology the use of means and medians for parametric and non-parametric data, respectively.

5. Table 2: “@=data are presented as mean ± SD” is added at the foot of the table, but the sign does not appear in the content of the table.

6. Line 186: P>0.999 is preferred over p=1

7. Line 189 to 196 Allelic discriminatin analysis, which genotype presents statistically significant differences for ATIC rs3821353, rs467399, SLC19A1 rs11702425, and GGH rs12681874?

8. Table 3. Some p values ​​are in bold, even though they are not statistically significant. Ejm: For the ATIC gene rs3821353 (over-dominant G/T p=0.07), ATIC rs16853834 (codominant T/T, p=0.89), DHFR rs1643657 (codominant A/G, p=0.21), SLC19A1 rs9977268 (codominant C /T, p=0.47) and SLC19A1 rs7279445 (codominant C/T, p=0.39 and T/T, p=0.67).

9. Table 3. Some p values ​​are statistically significant and are not in bold or (*); such is the case of: SLC19A1 rs11702425 (codominant C/T, p=0.02 and recessive T/T, p=0.02)

10. On line 281-282: I consider that care should be taken in pointing out that the haplotypes studied are associated with a high possibility of non-response to treatment, given that the data were not statistically significant.

Major revisions

1. Smoking is a variable that has been associated in various studies with the progression of RA disease, so it might be a good option to add the analysis of the effect of this variable on the response to treatment to Table 2. In addition, it is important to include how a smoking patient and a non-smoking patient were determined.

2. Methodology: in order to classify responders and non-responders, how many DAS28 evaluations were performed? At what time after the start of monotherapy treatment with MTX was the DAS28 obtained? Was the DAS28 evaluation time after treatment with MTX the same for all patients? I consider that the time in which the DAS28 is evaluated is a determining factor for whether or not a patient presents a good response. To prevent this data from being biased, an analysis could be included associating the time at which the DAS28 was evaluated. Include whether the patient is a responder or a non-responder.

3. It was determined that the polymorphisms are associated with the development of the disease, a fact that suggests that they are closely linked to the pathophysiology of RA; however, it is considered unlikely that these polymorphisms (exclusively linked to the drug pathway) contribute in any way to the development of RA disease. I believe that it is necessary for the authors to explain the association between these polymorphisms and the risk of developing RA disease, that is, it is important to explain how SNPs contribute to the pathophysiology of RA.

4. For the association of SNPs with drug efficacy, only 3 results showed importance in the study: ATIC rs3821353 (GT), DHFR rs10072026 (CT-CC) and SLC19A1 rs7279445 (CT-CC). However, very brief and light mention is made of these polymorphisms in the discussion. It is necessary to add and discuss these results. What is the effect of these polymorphisms? Is there an increase or decrease in enzymatic activity or greater or lesser expression of enzymes and/or transport proteins?

Author Response

Author’s Reply

Manuscript entitled: Pharmacogenomics of Methotrexate Pathway in Rheumatoid Arthritis Patients: Approach Toward Personalized Medicine

Dear respected reviewers,

We would like to express our sincere gratitude for the valuable comments and recommendations that helped to improve our manuscript. We have tried our best to modify the manuscript to fulfill all the comments. Kindly find below the point-by-point replies to your comments.

Reviewer 1 Minor Comments:

 Comment 1: Methodology: How was the adherence of the patients to the treatment evaluated during 3 months?  

Authors Reply: Thanks for the valuable remark. The method for evaluating adherence of the patients to the treatment was added to the text (Line 133-136).

Comment 2: Line 115: Since it is the first time that the abbreviation DAS28 is used, it is suggested to add the meaning.

Authors Reply: Thanks for this comment. DAS28 meaning was clarified in the text (Line 129-130).

Comment 3: Line 117 and 123: CRP add meaning of abbreviation. Idem CBC, ESR, S., RF.           

Authors Reply:  Full words were added before the abbreviations as suggested by the reviewer.

Comment 4: Methodology, line 151: It is mentioned that means and standard deviation were used; however, the results (table 2) also show medians. Please specify in the methodology the use of means and medians for parametric and non-parametric data, respectively.      

Authors Reply: Thanks for the valuable remark. We specified in the methodology the use of means and medians for parametric and non-parametric data, respectively.             

Comment 5: Table 2: “@=data are presented as mean ± SD” is added at the foot of the table, but the sign does not appear in the content of the table.

Authors Reply: Thanks for the valuable remark. The symbol was deleted, and we added mean and SD in the first row of the table.

Comment 6: Line 186: P>0.999 is preferred over p=1.

Authors Reply: Modified as suggested by the reviewer.

Comment 7: Line 189 to 196 Allelic discrimination analysis, which genotype presents statistically significant differences for ATIC rs3821353, rs467399, SLC19A1 rs11702425, and GGH rs12681874?

Authors Reply: Genotypes were added as requested by the reviewer.

Comment 8: Table 3. Some p values are in bold, even though they are not statistically significant. Ejm: For the ATIC gene rs3821353 (over-dominant G/T p=0.07), ATIC rs16853834 (codominant T/T, p=0.89), DHFR rs1643657 (codominant A/G, p=0.21), SLC19A1 rs9977268.

Authors Reply: Modified as suggested by the reviewer and deleted from text.

Comment 9: Table 3. Some p values are statistically significant and are not in bold or (*); such is the case of: SLC19A1 rs11702425 (codominant C/T, p=0.02 and recessive T/T, p=0.02).

Authors Reply: Modified as suggested by the reviewer and added to the text.

Comment 10: On line 281-282: I consider that care should be taken in pointing out that the haplotypes studied are associated with a high possibility of non-response to treatment, given that the data were not statistically significant.

Authors Reply: The reviewer suggested statement was added to highlight this observation.

Reviewer 1 Major Revisions:

Comment 1: Smoking is a variable that has been associated in various studies with the progression of RA disease, so it might be a good option to add the analysis of the effect of this variable on the response to treatment to Table 2. In addition, it is important to include how a smoking patient and a non-smoking patient were determined.

Authors Reply: Smoking as a variable was added to table 2. Also, description of smoking assessment was added to the methodology section (Line 127-128).

Comment 2: Methodology: in order to classify responders and non-responders, how many DAS28 evaluations were performed? At what time after the start of monotherapy treatment with MTX was the DAS28 obtained? Was the DAS28 evaluation time after treatment with MTX the same for all patients? I consider that the time in which the DAS28 is evaluated is a determining factor for whether or not a patient presents a good response. To prevent this data from being biased, an analysis could be included associating the time at which the DAS28 was evaluated. Include whether the patient is a responder or a non-responder.

Authors Reply: Thank you for this comment. Indeed, the DAS was evaluated for all patients at the same time interval from starting mono therapy. We highlighted the precise procedures encountered for DAS28 evaluation (Line 136-142)

Comment 3: It was determined that the polymorphisms are associated with the development of the disease, a fact that suggests that they are closely linked to the pathophysiology of RA; however, it is considered unlikely that these polymorphisms (exclusively linked to the drug pathway) contribute in any way to the development of RA disease. I believe that it is necessary for the authors to explain the association between these polymorphisms and the risk of developing RA disease, that is, it is important to explain how SNPs contribute to the pathophysiology of RA.

Authors Reply: Thank you for this comment. Kindly note that according to most of the published literature polymorphisms understudy are exclusively linked to the drug pathway as the reviewer mentioned. But with the current availability of new predictive software that analyze big data and can predict gene-disease novel associations, as VarElect (a cutting-edge variant election application for disease/phenotype-dependent gene variant prioritization via rapidly prioritizing genes that have been found to have variants according to selected disease/phenotype - gene associations https://ve.genecards.org/), we run a query composed of the genes understudy and specified rheumatoid as the selected disease/phenotype and we got the results attached below for possible disease association of these genes and RA. So, we decided to test this hypothesis experimentally and enrolled samples from normal controls in the study and compared their genotypes via allelic discrimination analysis and the results supported the initial hypothesis for an existing relation. As depicted from the figures below, the green score indicated the strength of the connection between the genes understudy and RA revealing that SLC19A1 and ATIC genes to be superior to the other two genes (details about the method of scoring are shown in the figure screenshot and available on the provided web link for the application). On the other hand, the orange score denoting the average disease-causing likelihood reflecting the principle that a variant in a gene with high mutation intolerance is more likely to be disease causing, pinpointed the 4 genes understudy to be implicated in causing RA with top score for DHFR followed by SLC19A1, ATIC, and GGH in descending order.

Finally, we totally agree with the reviewer that these results are novel and need further investigation in larger and different cohorts with different ethnicity to confirm it.

Comment 4: For the association of SNPs with drug efficacy, only 3 results showed the importance in the study: ATIC rs3821353 (GT), DHFR rs10072026 (CT-CC), and SLC19A1 rs7279445 (CT-CC). However, very brief and light mention is made of these polymorphisms in the discussion. It is necessary to add and discuss these results. What is the effect of these polymorphisms? Is there an increase or decrease in enzymatic activity or greater or lesser expression of enzymes and/or transport proteins?

Authors Reply: Thank you for this comment. Kindly note that we have added the requested details regarding the anticipated possible functional effects of these polymorphisms in the discussion section (Line 356-368).

Reviewer 2 Report

This is a very nice and interesting study of the effect of different SNPs of the genes involved in methotrexate metabolism in RA patients in a homogenous study group of the same ethnic origin.  The authors should try to improve their introduction explaining within the introduction and the abstract how they chose the specific SNPs they studied.

Author Response

Author’s Reply

Manuscript entitled: Pharmacogenomics of Methotrexate Pathway in Rheumatoid Arthritis Patients:

Approach Toward Personalized Medicine

Dear respected reviewers,

We would like to express our sincere gratitude for the valuable comments and recommendations that helped to improve our manuscript. We have tried our best to modify the manuscript to fulfill all the comments. Kindly find below the point-by-point replies to your comments.

Reviewer 2 Revisions:

 Comment 1: The authors should try to improve their introduction by explaining within the introduction and the abstract how they chose the specific SNPs they studied.

Authors Reply: Thank you for this comment. We added a section related to the specific chosen SNPs in the background section (Line 91-103). We also added a more detailed section to the methods clarifying the SNP selection process using the PharmGKB database with a new table 1B (Line 159-166).

Reviewer 3 Report

The authors in this great work describe some genes involeved in MTX pathway that could be helpful in clinical response to this DMARD.

The introduction is clearly presented with logic sequences.

In the methods please decribe the criteria of NON- RESPONDERS

Please specify why you choose this 10 genes in the methods

DO you have dta regarding the genes and disease activity and in smokers and in non smokers? Dod you faind any difference in men and women?

The figures are really clear and well descrinde in the results.

Author Response

Author’s Reply

Manuscript entitled: Pharmacogenomics of Methotrexate Pathway in Rheumatoid Arthritis Patients:

Approach Toward Personalized Medicine

Dear respected reviewers,

We would like to express our sincere gratitude for the valuable comments and recommendations that helped to improve our manuscript. We have tried our best to modify the manuscript to fulfill all the comments. Kindly find below the point-by-point replies to your comments.

 Reviewer 3 Revisions:

 Comment 1: In the methods, please describe the criteria of NON- RESPONDERS.

Authors Reply: Thank you for this comment. Criteria for both responders and non-responders were added (Line 142 to 146)

Comment 2: Please specify why you choose these 10 genes in the methods

Authors Reply: Thank you for this comment. We added a section to the methods clarifying the SNP selection process using the PharmGKB database with a new table 1B (Line 159-166).

Comment 3: Do you have data regarding the genes and disease activity in smokers and in non-smokers?

Authors Reply: According to the table below illustrating the association between different genotypes understudy and RA patients’ demographics and clinical characteristics, there was not any statistically significant relation between smoking or DAS28 with any of the SNPs under study.

Variables

V1

V2

V3

V4

V5

V6

V7

V8

V9

V10

Age

0.47

0.34

0.98

0.38

0.56

0.42

0.81

0.16

0.05

0..81

Gender

0.57

1

0.63

1

0.28

0.57

0.67

1

0.18

0.86

Family History

0.33

0.20

0.86

0.86

0.38

0.20

0.40

0.33

0.09

.015

BMI

0.54

0.68

0.95

0.21

0.13

0.46

0.39

0.89

0.46

0.62

Smoking

0.27

0.42

0.64

0.86

0.50

0.50

0.19

0.79

0.05

0.30

RA duration

0.19

0.62

0.28

0.30

0.71

0.28

0.39

0.03

0.87

0.99

TJC

0.20

0.04

0.55

0.83

0.73

0.42

0.34

0.60

0.09

0.59

SJC

0.85

0.32

0.18

0.97

0.54

0.27

0.11

0.12

0.71

0.83

VAS pain

0.37

0.34

0.87

0.45

0.25

0.88

0.39

0.19

0.24

0.99

Hb level

0.35

0.84

0.54

0.30

0.91

0.34

0.02

0.31

0.84

0.79

Plt count

0.40

0.97

0.43

0.11

0.91

0.51

0.63

0.92

0.13

0.55

TLC

0.40

0.47

0.69

0.89

0.84

0.35

0.61

0.30

0.83

0.42

ALT

0.85

0.72

0.12

0.20

0.26

0.48

0.26

0.45

0.44

0.28

AST

0.56

0.56

0.20

0.25

0.13

0.48

0.44

0.23

0.64

0.82

S. Creatinine

0.08

0.92

0.45

0.29

0.72

0.16

0.28

0.08

0.98

0.82

ESR

0.16

0.23

0.17

0.30

0.89

0.41

0.93

<0.01

0.97

0.28

CRP

<0.01

0.39

<0.01

0.37

0.48

0.49

0.89

0.49

0.72

0.53

RF

0.81

0.95

0.92

0.69

0.67

0.16

0.38

0.99

0.39

0.63

Anti-CCP

0.27

0.71

0.23

0.82

0.83

0.70

0.55

0.49

0.58

0.91

DAS-28

0.24

0.50

0.25

0.66

0.87

0.48

0.25

0.11

0.34

0.65

 † P value calculated using Chi Square/Fisher exact tests. For all other variables One Way ANOVA was used to compute the p value. V1: rs38213533 (ATIC), V2: rs4673990 (ATIC), V3: rs16853834 (ATIC), V4: rs10072026 (DHFR), V5: rs12517451 (DHFR), V6: rs1643657 (DHFR), V7: rs7279445 (SLC19A1), V8: rs11702425 (SLC19A1), V9: rs9977268 (SLC19A1), V10: rs12681874 (GGH).

Comment 4: Did you find any difference in men and women?

Authors Reply: According to the table above illustrating the association between different genotypes understudy and RA patients’ demographics and clinical characteristics, there were not any statistically significant relation between gender and any of the SNPs under study.
